# Circumstances and factors of sleep-related sudden infancy deaths in Japan

**Motoki Osawa** [1]*, **Yasuhiro Ueno**[2], **Noriaki Ikeda**[3], **Kazuya Ikematsu**[4], **Takuma Yamamoto**[5], **Wataru Irie**[6], **Shuji Kozawa**[7], **Hirokazu Kotani**[8], **Hideki Hamayasu**[8], **Takehiko Murase**[4], **Keita Shingu**[4], **Marie Sugimoto**[2], **Ryoko Nagao**[1], **Yu Kakimoto**[1]

1 Department of Forensic Medicine, Tokai University School of Medicine, Isehara, Kanagawa, Japan,
2 Department of Legal Medicine, Kobe University Graduate School of Medicine, Kobe, Hyogo, Japan,
3 Department of Forensic Pathology and Sciences, Graduate School of Medical Science, Kyushu University, Fukuoka, Japan, 4 Department of Forensic Pathology and Science, Graduate School of Biomedical Sciences Nagasaki University, Nagasaki, Japan, 5 Department of Legal Medicine, Hyogo College of Medicine, Nishinomiya, Hyogo, Japan, 6 Department of Legal Medicine, Kitasato University School of Medicine, Sagamihara, Kanagawa, Japan, 7 Department of Forensic Medicine and Sciences, Mie University School of Medicine, Tsu, Mie, Japan, 8 Department of Forensic Medicine and Molecular Pathology, Kyoto University Graduate School of Medicine, Kyoto, Japan

* osawa@is.icc.u-tokai.ac.jp

**Data Availability Statement:** All relevant data are within the manuscript and its Supporting Information files.

## Abstract

### Background

Sudden unexpected death in infancy (SUDI) comprises both natural and unnatural causes of death. However, few epidemiological surveys have investigated SUDI in Japan.

### Objective

This retrospective study was conducted to investigate the latest trends of circumstances and risk factors of SUDI cases in which collapse occurred during sleep.

### Methods

Forensic pathology sections from eight universities participated in the selection of subjects from 2013 to 2018. Data obtained from the checklist form were analyzed based on information at postmortem.

### Results

There were 259 SUDI cases consisting of 145 male infants and 114 female infants with a mean birth weight of 2888 ± 553 and 2750 ± 370 g, respectively. Deaths most frequently occurred among infants at 1 month of age (18%). According to population data as the control, the odds ratio (95% confidence interval) of mother's age ≤19 years was 11.1 (6.9–17.7) compared with ages 30–39. The odds ratio for the fourth- and later born infants was 5.2 (3.4–7.9) compared with the frequency of first-born infants. The most frequent time of day for discovery was between 7 and 8 o'clock, and the time difference from the last seen alive was a mean of 4.1 h. Co-sleeping was recorded for 61%, and the prone position was found

**Funding:** M.O. received Grants-in-Aid from the Health, Labour and Welfare Sciences Research Grants, Japan (grant no. H29-Sukoyaka-001).

**Competing interests:** The authors have declared no competing interests exist.

for 40% of cases at discovery. Mother's smoking habit exhibited an odds ratio of 4.5 (2.9–5.8).

## Conclusion

This study confirmed the trends that have been observed for sudden infant death syndrome; particularly, very high odds ratios were evident for teenage mothers and later birth order in comparison with those in other developed countries. Neglect was suspected in some cases of the prolonged time to discovery of unreactive infants. To our knowledge, this is the first report of an extensive survey of SUDI during sleep in Japan.

## Introduction

Sudden infant death syndrome (SIDS) is the possible cause-of-death for sudden infant death during sleep, in which all known identifiable conditions that might engender sudden and unexpected death must be excluded by postmortem examinations. However, several pathologists have changed their diagnostic preferences since 2004 primarily because of the difficult distinction of SIDS from accidental asphyxia or natural diseases such as arrhythmias and metabolic disorders. [1–3] Reluctance to use the term has decreased the number globally over the years. [4–6] In Japan, around 0.4 per 1,000 livebirths (LBs) of annual SIDS rate was recorded in the 1990s [7], but recent diagnostic numbers have decreased to fewer than 0.1. [8]

Currently, another broad term has become popular as an alternative to a final SIDS decision, i.e., sudden unexpected death in infancy (SUDI) or sudden unexpected infant death (SUID) in an apparently healthy infant presenting at 7–365 days of age. Although SUDI/SUID originally has been used as an umbrella term for the initial presentation of explained or unexplained infant deaths, it is interpreted to include several categories such as SIDS (R95), ill-defined and unknown cause of mortality (R99), where the investigation, death scene examination, autopsy findings were limited or incomplete. Unintentional threat to breathing (W75-W84) to include sudden death in infants, with bed/sleep surface sharing, soft bedding, or prone sleep, without adequate evidence for airway obstruction or chest wall compression, are insufficient to certify a death as due to asphyxia. [9] Shapiro-Mendoza et al. [10] demonstrate a decision-making algorithm for assigning SUID case registry.

In recent years, a protocol of investigation items has been standardized. [11,12] For instance, in the U.S., the Centers for Disease Control and Prevention published guidelines and a reporting form, which is designated as the SUID Investigation Reporting Form. [13] In Japan, a list of items to be investigated is used as a similar checklist form in cases of infant death. [14] A system is in operation for clinicians and pathologists to ascertain circumstances and to investigate background factors. Information from death scene investigation (DSI) acquired by experts is indispensable to fill out the form. [15] Furthermore, the maternity passbook, which records information about the mother and the child during pregnancy as well as after childbirth, is beneficial. We previously analyzed forensic autopsy cases of sudden infant deaths after vaccination using this passbook. [16]

While the childcare environment differs according to region and time, there are few epidemiological data describing infant deaths during sleep in Japan. [17] Takatsu et al. [18] exhibited the trends of SUID cases from 1982 to 2006. This population-based retrospective study was conducted to investigate the latest trends of sleep-related SUDI cases using the checklist form including the maternity passbook at multiple centers.

## Methods

Sleep-related SUDI cases were recruited from autopsy files for the period of 6 years from 2013 to 2018 to obtain sufficient number of cases for adequate statistical power. Inclusion criteria for cases were age not less than 1 week and not more than 12 months, and the collapse occurring during sleep in an unexpected manner. Based on the cause and manner of death, infant deaths were grouped as follows: (1) infants who died of SIDS, (2) infants who died of other natural diseases, (3) infants who died of accidental injuries, (4) infants who died of non-accidental injuries, and (5) infants with undetermined manner of death classified under R99, W74-84. [19] In this study, we selected cases of groups (1) and (5) and those of suspected accidental suffocation during sleep.

Postmortem examinations included histology, toxicology, biochemistry, virology, and bacteriology. [20,21] Tests for assessing inherited metabolic disorders were conducted nationwide in the routine examination for newborns. [22] Genetic testing for arrhythmic disorders was also conducted for cases examined in this study. [23] Data used for analysis consisted of DSI information, therapeutic information in emergency care, and maternity passbook. The checklist form, consisting of 41 items, was filled in initially by each center. Then the lists were transferred to one site to confirm unclear issues and aggregate the data.

The forensic pathology sections of the following eight universities participated in this study: Kitasato University School of Medicine, Mie University School of Medicine, Kyoto University Graduate School of Medicine, Hyogo College of Medicine, Kobe University Graduate School of Medicine, Graduate School of Medical Science Kyushu University, Graduate School of Biomedical Sciences Nagasaki University, and Tokai University School of Medicine. The areas of these facilities cover six prefectures without regional bias in the country. Further, the area of coverage for these institutions comprised approximately 14% of where the entire population resides, and this percentage was applied to approximate the annual rate per 1000 LBs in the area served by the institutions as 14 per cent of the national population birth rate. Every sudden infant death had been autopsied, but there could have been some deaths that had not received autopsy outside domestic forensic service centers in Japan, whose exact number remained unknown. The principal investigator obtained approval for this retrospective study from the Institutional Review Board for Clinical Research, Tokai University. This study was also approved by the respective ethical committees of the faculties as a collaborative study. All data were fully anonymized before the analyzing investigators accessed them. The study protocol was disclosed to the public at the website.

The number of 263 cases were originally registered to this project. We checked each candidate case carefully at a meeting, and selected subjects in which terminal events remained at speculation irrespective of the diagnosis in the death certificate. Among them, 4 cases were eliminated from the reasons of not found in sleep environment ($n = 3$) and expected course of terminal heart failure ($n = 1$). The original causes of death ($n = 259$) were SIDS in 94 cases (36%), undetermined (R99) in 75 cases (29%), potential asphyxia in bed in 51 cases (20%), and SUDI with other natural causes in 39 cases (15%) where the causes were not fatal in itself and there was co-sleeping as a risk factor. Suspected causes of asphyxia were supposed to be due to accidental overlay and choking of milk in bed. Inflammation of the airway, including bronchitis, accounted for 22 cases, comprising the largest group among "others." Pathological findings do not always reflect the terminal event. Pathologists tend to favor any apparent histological findings as the cause of death whether it was fatal or not. Despite of such histological evidence, the pathologists reconsidered that these cases might also be regarded as sleep-related SUDI because of co-sleeping, in which coexistent factors may have served as contributors in causing death.

Since we had too small number of non-SUDI cases that were suitable for the control, we used local population data approximated from national population data as a substitute for the control cases in a comparison with the present data. The statistical data of LBs recorded during 2013–2018 in the Japanese population at the National Institute of Population and Social Security Research were used as the control. [8] Prevalence information of regular tobacco consumption was available at Japan Tobaco Inc. as a questionnaire survey performed in 2016 (https://www.jti.co.jp/investors/press_releases/2016/0728_01.html). The numbers of smokers and nonsmokers in the 20s and 30s of female volunteers were used for the control.

Logistic regression analyses based on population data were performed to determine the associations of independent variables as estimated using the odds ratio and confidence interval (CI). For significant differences, the *p* value was calculated using chi-square tests. Statistical analyses were conducted using BellCurve in Excel, ver. 3.20 (Social Survey Research Information Co., Tokyo, Japan).

## Results

A total of 259 cases were collected at multiple centers for the 6-year period. The circumstances and factors were investigated using the checklist form filled with information at postmortem.

The annual frequency of SUDI during sleep was estimated to approximately 0.31 per 1000 LBs. However, the value could be slightly underestimated because of the possibility that some cases could have been out of management by these facilities.

Table 1 presents the number of subjects according to sex, birth weight, gestation week, maternal age, parity and maternal smoking habit along with the population data. Table 2 summarizes the odds ratios (95% CI) and *p* values in terms of the related factors.

### Age

Fig 1 shows the age distribution at the time of death. It was observed that deaths most frequently occurred in infants at 1 month of age, consisting of 45 cases (18%). The number was found to decrease with age. Deaths occurring within 6 months after birth accounted for 180 cases (72%).

### Birth weight and gestation weeks

The mean (± S.D.) birth weight of SUDI infants was 2885 ± 556 g for male subjects (n = 137) and 2763 ± 466 g for female subjects (n = 108). These birth weights were lower by 191 g (6%) and 227 g (8%), respectively, than the national mean birth weights of 3076 g for males and 2990 g for females recorded in 2017. The incidence of low birth weight infants was significantly higher than the control group in both sexes. For low birth weight infants, the odds ratio of over 2.0 was observed in both the male and female groups. Infants of premature birth were found to be 2.6 times more likely to die from SUDI than those of mature birth.

### Maternal age and birth order

The odds ratio of incidence of infant death of mothers whose age ≤19 years was the highest at 11.1 compared with mothers aged 30–39 years, and that for mothers aged 20–29 years was 2.1, which showed significant differences. This finding indicated that mothers of a younger age, especially teenage, should be considered as the most important risk factor for the occurrence of SUDI during sleep.

In terms of the birth-order distribution, there were 31% of first-born infants, 38% of second-born infants, 19% of third-born infants, 8% of fourth-born infants, 2% (6 cases) of fifth-

**Table 1. Number of SUDI subjects and LBs.**

| Factors | No. (%) of SUDI | No. (%) of LBs* | Approx. annual rate per 1000 LBs** |
|---|---|---|---|
| Sex | | | |
| Male | 145 (56%) | 3,015,822 (51%) | 0.34 |
| Female | 114 (44%) | 2,864,653 (49%) | 0.28 |
| Total | 259 (100%) | 5,880,475 (100%) | 0.31 |
| Birth weight: Male | | | |
| < 2,500 | 21 (15%) | 252,678 (8%) | 0.59 |
| ≥ 2,500 | 116 (85%) | 2,762,747 (92%) | 0.30 |
| Total | 137 (100%) | 3,015,425 (100%) | 0.32 |
| Birth weight: Female | | | |
| < 2,500 | 23 (21%) | 304,624 (11%) | 0.54 |
| ≥ 2,500 | 85 (79%) | 2,559,683 (89%) | 0.24 |
| Total | 108 (100%) | 2,864,307 (100%) | 0.27 |
| Gestation week | | | |
| < 37 | 31 (13%) | 324,829 (6%) | 0.68 |
| ≥ 37 | 203 (87%) | 5,546,995 (94%) | 0.26 |
| Total | 234 (100%) | 5,871,824 (100%) | 0.28 |
| Maternal age at delivery (years) | | | |
| ≤ 19 | 21 (9%) | 67,675 (1%) | 2.22 |
| 20–24 | 63 (26%) | 500,757 (9%) | 0.90 |
| 25–29 | 58 (24%) | 1,538,223 (26%) | 0.27 |
| 30–34 | 53 (22%) | 2,124,833 (36%) | 0.18 |
| 35–39 | 43 (17%) | 1,335,169 (23%) | 0.23 |
| 40–44 | 7 (3%) | 305,543 (5%) | 0.16 |
| ≥ 45 | 0 | 8,268 (0%) | |
| Total | 245 (100%) | 5,880,468 (100%) | 0.30 |
| Parity | | | |
| 1 | 77 (31%) | 2,745,441 (47%) | 0.20 |
| 2 | 93 (38%) | 2,145,351 (37%) | 0.31 |
| 3 | 46 (19%) | 760,283 (13%) | 0.43 |
| 4 | 20 (8%) | 157,536 (3%) | 0.91 |
| ≥ 5 | 10 (4%) | 50,823 (1%) | 1.41 |
| Total | 246 (100%) | 5,859,434 (100%) | 0.30 |
| Maternal Smoking habit | | | |
| Non-smoker | 106 (66%) | 1,983 (89%) | - |
| Smoker | 55 (34%) | 254 (11%) | - |
| Total | 161 (100%) | 2,237 (100%) | |

*: Japanese population data represent the sum of LBs during 2013 to 2018.

**: The annual rate is calculated by no. of SUDI / (no. of national LB × 0.14)

born infants, 1% (3 cases) of sixth-born infants, and 0.4% (1 case) of seventh-born infant. The odds ratio to the fatal frequency among the first-born infants clearly indicated that later birth order constituted an important risk factor. Moreover, as shown in Fig 1, there were more first-born infant deaths within 2 months in age (38/92) than those after 3 months (39/154) ($p < 0.001$).

**Table 2. Odds ratios (95% CI) and *p* values of SUDI cases to population data.**

| Factors | Odds ratio | *p* value |
|---|---|---|
| Sex | | |
| Male | 1.2 (0.9–1.5) | *p* = .13 |
| Female (ref. group) | 1.0 | |
| Birth weight; Male | | |
| < 2,500 | 2.0 (1.3–3.2) | *p* < .01 |
| ≥ 2,500 (ref. group) | 1.0 | |
| Birth weight; Female | | |
| < 2,500 | 2.3 (1.4–3.6) | *p* < .001 |
| ≥ 2,500 (ref. group) | 1.0 | |
| Gestation week | | |
| < 37 | 2.6 (1.8–3.8) | *p* < .001 |
| ≥ 37 (ref. group) | 1.0 | |
| Maternal age at delivery (years) | | |
| ≤ 19 | 11.1 (6.9–17.7) | *p* < .001 |
| 20–29 | 2.1 (1.6–2.8) | *p* < .001 |
| 30–39 (ref. group) | 1.0 | |
| ≥ 40 | 0.8 (0.4–1.7) | *p* = .56 |
| Parity | | |
| 1 (ref. group) | 1.0 | |
| 2–3 | 1.7 (1.3–2.3) | *p* < .001 |
| ≥ 4 | 5.1 (3.4–7.8) | *p* < .001 |
| Maternal smoking habit | | |
| Non-smoker (ref. group) | 1.0 | |
| Smoker | 4.1 (2.9–5.8) | *p* < .001 |

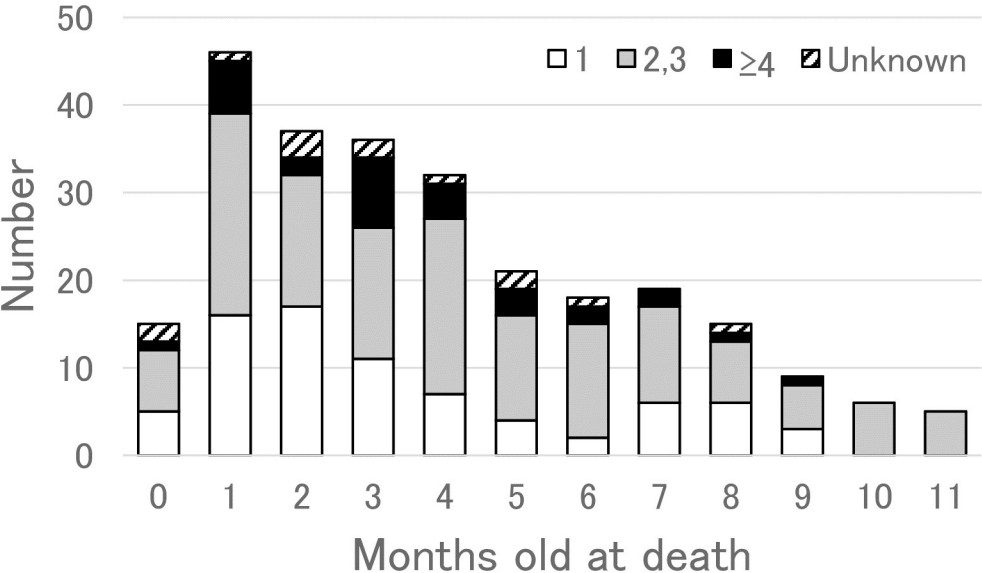

**Fig 1. Age distribution of SUDI infants during sleep (*n* = 259) examined in this study.** Key indicates the birth order of infants, in which the column is divided into four groups of the first-born infants (blank), the second- and third-born ones (gray), more than the fourth-born ones (black), and unknown ones (diagonal).

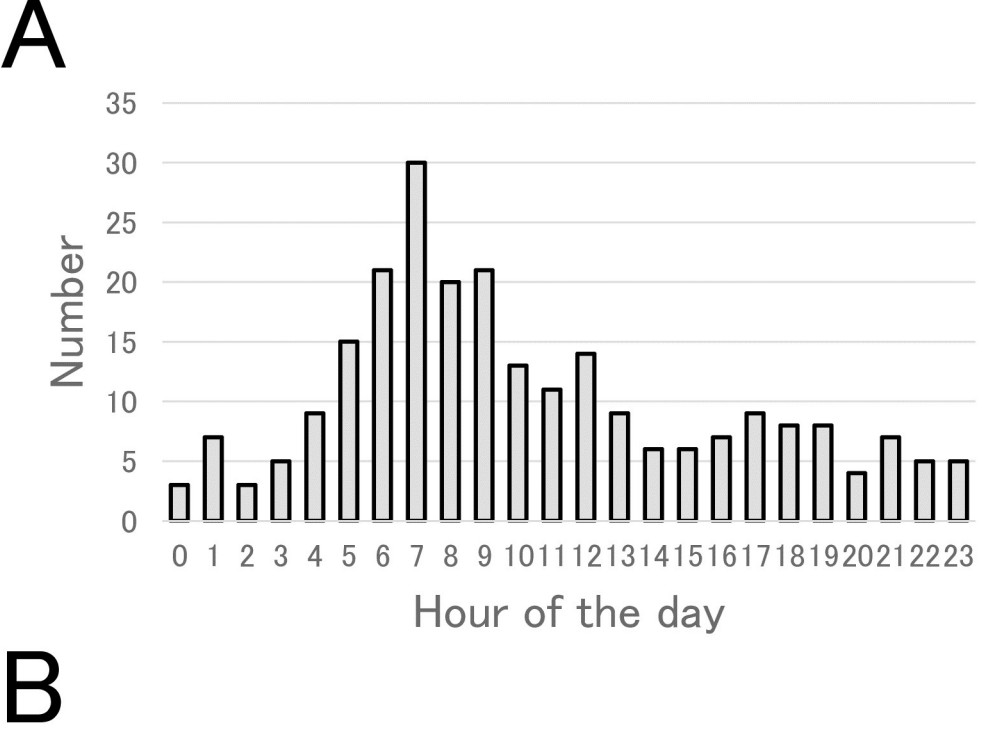

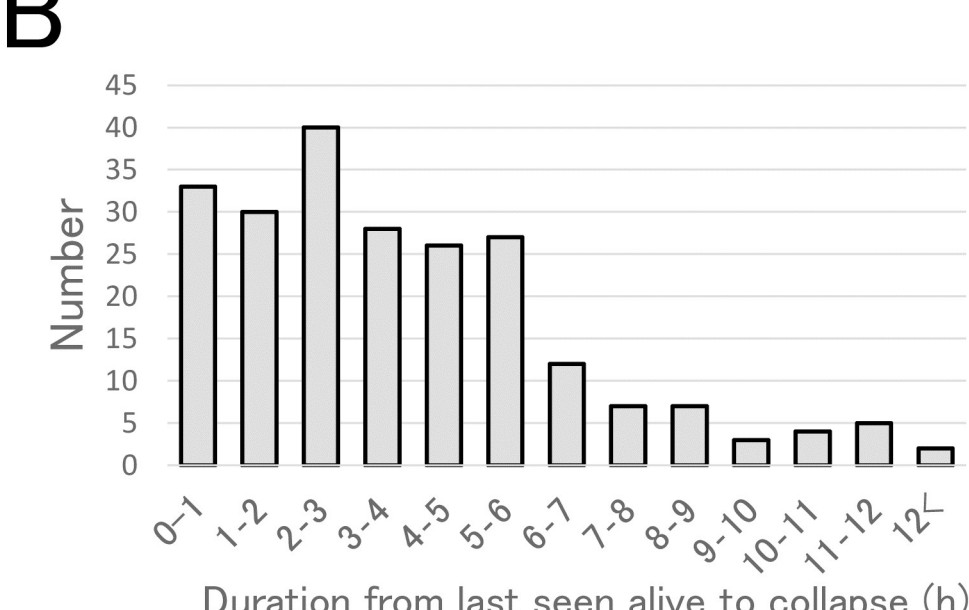

**Fig 2. Time of discovery.** A. Distribution of the time of the day at which the first responder found an unresponsive infant (*n* = 246); B. the time difference between the time the infant was last seen alive and first found deceased (*n* = 222).

## Time of discovery and sleeping position

Fig 2A shows the distribution of the time of day when an unresponsive infant was found. There were 30 cases (12%) found between 7 and 8 o'clock a.m., which was the most frequent time. A large peak was evident between 6 and 9 o'clock in the morning.

**Table 3. Sleep position at the scene.**

| Position | Total (0–11 month) | 0–2 month | 3–11 month |
|---|---|---|---|
| Supine | 117 (52%) | 60 (72%) | 57 (40%) |
| Prone | 89 (40%) | 16 (19%) | 73 (52%) |
| Side | 16 (7%) | 7 (9%) | 9 (6%) |
| Others | 2 (1%) | 0 | 2 (1%) |
| Total | 224 (100%) | 83 (100%) | 141 (100%) |

Fig 2B depicts that the duration between the last time the infant was found alive and the time of discovery of being unresponsive ($n = 222$), which varied widely from approximately 10 min to 13 h. The mean duration was 4.1±2.7 h. The collapse was discovered within 6 h in the majority of cases ($n = 184$, 83%).

The first responder ($n = 252$), who discovered the unresponsive infant, was the mother in 188 cases (75%), followed by the father in 49 cases (19%), a grandmother in 8 cases (3%), a childminder in 2 cases (1%), and others in 6 cases (2%).

Co-sleeping was recorded for 143 cases (61%) among a total of 230 available cases. Table 3 presents the child sleeping position when the collapse was discovered. The prone position in late SUDI infants accounted for 40% of cases, and there were 19% of cases with the prone position even in 0 to 2-month-old subjects who cannot roll over.

## Maternal smoking habit

The descriptions in the maternity passbook entries are considered to reflect the smoking habits before and during the early phase of pregnancy. We attempted to obtain the smoking rate of the mothers of SUDI cases. A significant risk of SUDI was evident with an odds ratio of 4.5 compared with the general rate. The mean number of cigarettes was 11 cigarettes/day ($n = 30$).

## Discussion

Since we had a small number of control cases that were suitable for a case-control study, the national population data was used as a substitute for the reference. In addition, there was a limitation that data were not fully available for each item because of the retrospective approach. Whilst the analysis might be accompanied by some imprecision, high odd ratios were evident in low birth weight, premature birth, teenage mothers, later birth order infants and maternal smoking habit.

These results of the present investigation of sleep-related SUDI cases were consistent with risk factors such the smoking habit of parents in the large epidemiological surveys of SIDS. [24] However, some differences were evident. The peak age of death is generally 2 months in SIDS surveys, [4,25] but that among the present SUDI infants was 1 month of age. We thought that the difference should be caused by that a particularly higher risk was evident among teenage mothers than that found in an earlier study [26], and that more first-born infants had died during 0–2 months of age.

The most frequent birth order associated with infant death due to SUDI during sleep was the second birth order. Blair et al. [27] reported that SIDS was most frequent among first-born children in the UK, although it was earlier presumed to be frequent in large families. Data from Taiwan indicate that the first-, second-, and third- and later born children account for 36%, 40%, and 24% of SIDS, respectively. [28] The distribution in the present study was more similar to that reported in Taiwan.

Traditional bedding of cotton mat, known as futon, on the floor is common in Japan. Therefore, it is more appropriate to use the term co-sleeping (sharing a sleeping surface) than bed-sharing. Such co-sleeping is a common style of sleep. Tokutake et al. [17] reported that 84% of mothers practice co-sleeping, of whom half also practice breastfeeding. The father was found to be the first responder in up to 20% of cases, and in most of these cases the father also co-slept and discovered the infant death upon awakening. The risk of SIDS among infants who co-sleep was found to be significantly high in several earlier studies. [29,30] In addition, the higher incidence could be related to the inclusion of infants who had associated natural disease as cause of death but included in this study if there was co-sleeping. Nevertheless, the effects of co-sleeping on the occurrence of SUDI, if any, could not be evaluated in this study because of a variety of co-sleeping styles and the absence of good control subjects.

It is a traditional practice in Japan to lay infants in the supine position. However, 40% of infants were found in the prone position, of which frequency was higher than that reported in an earlier study. [17] Li et al. [31] reported that 60% of SIDS infants were found in the prone position in the United States. It is possible that turning over by infants during sleep is a causal factor. However, approximately 28% of 0 to 2-month-old infants who were unable to turn over were found in the prone and side positions. They might have been placed in the prone position or been breastfed during co-sleeping, but the original position was not recorded sufficiently at DSI.

A striking finding was the prolonged time to discovery. In nine cases, it spent more than 10 hours to find the unreactive infants. No apparent infanticide was involved in all subjects, however neglect by parents was suspected in a couple of cases from circumstances at DSI. We think that the time difference should be an important indicator to suspect careless infant rearing.

A relationship between the occurrence of SIDS and the smoking habit of parents has been found in Japan. [32] In the present investigation, the incidence of pregnant mother's smoking among SUDI cases was 34% despite of the limited number of subjects. This incidence in the general female population was reported as 11%, which also resulted in the high odds ratio of 4.5 in this study. According to Anderson et al., [33] the incidence of SUDI more than doubles when a parent is smoking during the period of pregnancy. The odds ratio increases along with the number of cigarettes up to 20. It is evident that infants co-sleeping with someone who smokes exhibit the highest risk for SUDI. [34]

Pasquale-Styles et al. [35] reported that asphyxia and suffocation occur more than presumed in many situations such as bed-sharing, overlay, wedging, prone position, obstruction of the nose and mouth, and coverage of the head. Postmortem findings alone are not generally sufficient to explain the cause of these deaths. Consequently, the diagnoses often lack consistency. [3,4] To avoid preventable any types of SUDI, including SIDS, such various causes of accidental suffocation, and unexplained causes, it is important to identify high risk factors from the study based on the wide variety of cases. [18] In addition, there exists a difficulty of the current situation in Japan, particularly in DSI that is performed by police officers who are not well trained. Although DSI was performed by the police for all present cases, not all items were optimal, particularly, for the sleep environment such as sleep surface, wrapping, and clothing Garstang et al. [25] indicated that police-led DSI does not comply with practical information. After a new law related to child health was enacted in 2018, child death reviews will be introduced to the society in the near future. These reviews in combination with multiple agencies will be helpful in investigating the sleeping environments of infants in detail.

In conclusion, we conducted an effective epidemiological analysis of sleep-related SUDI using the checklist form. This approach has revealed the present critical features prevailing in the country. This report displayed the latest trends of SUDI in Japan.

## Supporting information

**S1 Data.**
(XLSX)

## Acknowledgments

We thank Tomohiro Nozima for his extensive support like inputting the large data and mailing.

## Author Contributions

**Conceptualization:** Motoki Osawa.

**Data curation:** Yasuhiro Ueno, Noriaki Ikeda, Kazuya Ikematsu, Takuma Yamamoto, Wataru Irie, Shuji Kozawa, Hirokazu Kotani.

**Formal analysis:** Hideki Hamayasu, Takehiko Murase, Keita Shingu, Marie Sugimoto.

**Funding acquisition:** Motoki Osawa.

**Methodology:** Ryoko Nagao.

**Validation:** Yu Kakimoto.

**Writing – original draft:** Motoki Osawa.

**Writing – review & editing:** Takuma Yamamoto, Hirokazu Kotani, Yu Kakimoto.

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
