## [Decision Letter · Decision Letter 0]

5 Jun 2020

PONE-D-20-12699

Circumstances and factors of sleep-related sudden infant deaths in Japan

PLOS ONE

Dear Dr. Osawa,

Thank you for submitting your manuscript to PLOS ONE. After careful consideration, we feel that it has merit but does not fully meet PLOS ONE’s publication criteria as it currently stands. Therefore, we invite you to submit a revised version of the manuscript that addresses the points raised during the review process from the reviewers.

We look forward to receiving your revised manuscript.

Kind regards,

Ju Lee Oei

Academic Editor

PLOS ONE

Journal Requirements:

2. In the ethics statement in the manuscript and in the online submission form, please provide additional information about the patient records used in your retrospective study. Specifically, please ensure that you have discussed whether all data were fully anonymized before you accessed them and/or whether the IRB or ethics committee waived the requirement for informed consent. If patients provided informed written consent to have data from their medical records used in research, please include this information.

Reviewers' comments:

Reviewer's Responses to Questions

**Comments to the Author**

1. Is the manuscript technically sound, and do the data support the conclusions?

Reviewer #1: Partly

Reviewer #2: Partly

2. Has the statistical analysis been performed appropriately and rigorously? 

Reviewer #1: No

Reviewer #2: No

3. Have the authors made all data underlying the findings in their manuscript fully available?

Reviewer #1: No

Reviewer #2: Yes

4. Is the manuscript presented in an intelligible fashion and written in standard English?

Reviewer #1: Yes

Reviewer #2: Yes

5. Review Comments to the Author

Reviewer #1: This paper describes the recent status of sudden unexpected death in infancy (SUDI) using the data obtained by a postmortem check in Japan. The study design was a retrospective case-control method. The authors analyzed 259 SUDI cases during 2013-2018 registered in 8 different universities. They found that SUDI most frequently occurred among infants at 1 month of age and significant associations with the infants born to mothers at their teenage, later born infants, and a smoking habit of a mother. They also mentioned possible associations with co-sleeping and prone positioning during sleep. Thus, the study is very important, however, the results from this study are weakened due to the lack of appropriate control subjects. Furthermore, the eligibility and selection criteria of sleep-related SUDI infants among all SUDI infants were not clear in the manuscript.

Major comments.

1 Clarify the relationship between SUDI and sleep-related SUDI in Abstract and Manuscript.

2 Method line 89. Explain why the study period of 2013-2018 was selected.

3 Method lone 90-96 and 120-129. Explain the selection and classification methods of the study subjects precisely. Not clear even how many SUDI were registered totally. When and who decided? How to get a final agreement among investigators? Any audit by the third party? This is the most important point of this study.

4 Method line 127. Is this stipulation feasible? Need more explanation.

5 Method line 111 and 130-132. If this study covered only 14% of the total population in Japan, it would be hard to say “population based” (line 40). Rather say “regional investigation”.

6 Method line 130-132. It would be better to use the population background of the prefectures where 8 universities belong to.

7 Results Table 2 line 162-163. Were these odds ratios adjusted against the background distributions of the control?

8 Results the same as above. Explain why the early group consists of 0-2 months of age.

9 Discussion. Summarize key results without repeating the data already mentioned in Results.

10 Discussion. Better to make a paragraph of limitations of the study with potential bias or imprecision.

11 Discussion line 250-252. Be careful about this interpretation because of the lack of control subjects.

Minor comments

1 Discussion lines 279 and 286. The authors said the DSI was not reliable while the checklist was an official, very confusing description.

2 Better to show a flow diagram of subjects.

Reviewer #2: This study seeks to study sleep related SUDI in Japanese infants and has successfully collected data from a large area in Japan over 6 years.

The authors have attempted to look at known risk factors from other preceding studies but the study could be improved with further details on social and family demographics, maternal education, breastfeeding status if available. Some of the objectives of this study could be to look at prevention of sleep related SUDI and improving investigation and documentation of DSI in Japan.

Some more specific comments :

Line 61. it might be easier to make a international comparison by giving rates instead of numbers per year, e.g.if rates of SIDS are given as 0.15 per 1000 livebirths

Line 73. Good that Japan has a SIDS investigation form - is it used for by the death scene investigation team or forensic pathologists? The authors might want to mention that it is similar to CDC SUDI investigation form since it is in Japanese.

Line 79. It would be interesting as an introduction to mention incidence of SUDI in Japan (0.4(?) per 1000 livebirths from Taylor et al's study) and any findings of sleep related SUDI from previous studies such as ref 17 or from using the DSI form, since that is the authors' major focus, rather than a mention of vaccination related SUDI

Line 88. The authors' case definition included infants 0-365 days. According to most definitions, unexpected deaths in infants under 7 days of age are excluded from the SUDI category, and instead have been termed “sudden unexpected early neonatal death (SUEND)”. It would be interesting to know how many infants in the study were actually below 7 days of age to see if it would impact the findings related to early or late SUDI.

Line 134 - alignment of words (for editor)

Line 157, do the authors mean the later group is in where death occurred at or after 3 months of age .. . What about the infants of 2 months plus to 3 months of age if the later group is AFTER 3 months of age and Early group is within 2 months of age (<= 2 months old). Do the authors mean "below 3 months of age" ?

From the existing literature, the age groups are usually neonatal SUDI (i.e. below 28 days , approximately 1 month of age) and after the neonatal period. Is it an empirical decision or to take into account the preterm babies ?

Line 159 Table 1. "SUDI" is used in the text but "SUID" is used in the Table, suggest to use one term only for consistency. Suggest to add 'livebirths' to become "Approx annual (incidence) rate per 1000 livebirths(LB) "

Line 179 - Is it the annual incidence rate of SUDI in low birth weight infants being significantly higher rather than the "percentage", the latter which is not shown in table 1

Line 223 Table 3 - typo Supine

Statistical analysis :

Please excuse me if my understanding of the analysis is incorrect ...

Table 1 -

1.the annual rate per 1000 LB for male infants would be [145*1000/no. of LB (in the 6 years) ] divided by 6 = 0.008 and so forth for the other variables. It may also not be necessary to have an annual incidence but just an overall incidence over the 6 years since the annual rate is very low

2. Line 190 and 191 - I am not sure if the number of early SUDI in one age group compared to early SUDI in the rest of the population can be used in chi squared calculation against total SUDI population- should it instead be compared with non event population in the same age group ?

Line 255. The effects of co-sleeping could not be evaluated but perhaps more details if available could be described under findings? Was there any related to wedging, inadvertent suffocation especially in co-sleeping cases ? Or to mention Line 280 ..."not well trained in documentation".

Discussion - Some comment re missing data would be helpful , 14 cases missing in gender , 25 cases in gestation and more than half cases had missing data on maternal smoking - how it may affect the reliability of the data esp with regards to smoking , although this is a known risk factor in sleep related SUDI. Sleeping position was missing in 35 cases. These are understandable given the retrospective nature of the study - so a comment may be useful to the reader

Line 238 - Could the authors comment if the peak age of death at one month is related to the age group being studied to be 0-12 months of age as compared to other SUDI studies where the study group is from 2 to 12 months ?

Prevention issues - of the risk groups are not discussed much . Any comments on how the babies of later birth order might be at risk ?

Is there a concern about neglect if the duration of an infant seen alive is more than 5-6 hours for those found in the morning, and more than 2-3 hours for those found in the afternoon (since the parents or childminder would be awake then)? Is there a likely delay in reporting due to possible infanticide/ negligence in those reporting that last seen alive was 8 hours ? Or is it a non carer who was reporting his duration? Lack of this data could be a discussion point in your study re improvement in investigation by the social workers or police.

Issues related to DSI itself could be a discussion point on quality improvement in documentation and investigative process, were there any cases which could have been missed infanticide ?

6. PLOS authors have the option to publish the peer review history of their article (what does this mean?). If published, this will include your full peer review and any attached files.

Reviewer #1: No

Reviewer #2: Yes: Irene Guat Sim CHEAH

---

## [Author Response · Author response to Decision Letter 0]

8 Jul 2020

The detailed review of our article is appreciated. The comments by the reviewer have been helpful in allowing us to revise the manuscript. The authors have attempted to address the questions raised as separate pages. According to the raised comments, the manuscript has been rewritten extensively to a revised version. Alterations are indicated as track changes in the revised manuscript.

---

## [Decision Letter · Decision Letter 1]

22 Jul 2020

PONE-D-20-12699R1

Circumstances and factors of sudden infant deaths during sleep in Japan

PLOS ONE

Dear Dr. Osawa,

Thank you for submitting your manuscript to PLOS ONE. After careful consideration, we feel that it has merit but does not fully meet PLOS ONE’s publication criteria as it currently stands. Therefore, we invite you to submit a revised version of the manuscript that addresses the points raised during the review process.

The comments from reviewer 2 are minor and it would help greatly if the authors could address them. 

We look forward to receiving your revised manuscript.

Kind regards,

Ju Lee Oei

Academic Editor

PLOS ONE

Reviewers' comments:

Reviewer's Responses to Questions

**Comments to the Author**

1. If the authors have adequately addressed your comments raised in a previous round of review and you feel that this manuscript is now acceptable for publication, you may indicate that here to bypass the “Comments to the Author” section, enter your conflict of interest statement in the “Confidential to Editor” section, and submit your "Accept" recommendation.

Reviewer #1: All comments have been addressed

Reviewer #2: (No Response)

2. Is the manuscript technically sound, and do the data support the conclusions?

Reviewer #1: Yes

Reviewer #2: Partly

3. Has the statistical analysis been performed appropriately and rigorously? 

Reviewer #1: Yes

Reviewer #2: I Don't Know

4. Have the authors made all data underlying the findings in their manuscript fully available?

Reviewer #1: Yes

Reviewer #2: No

5. Is the manuscript presented in an intelligible fashion and written in standard English?

Reviewer #1: Yes

Reviewer #2: No

6. Review Comments to the Author

Reviewer #1: The authors have answered all comments from the reviewer appropriately. No further comment to the authors.

Reviewer #2: 1. I would suggest to authors if removing the words "during sleep" and from the objective would make the study different? SUDI by definition happens during sleep. It would make the methodology wording less confusing

2. The methodology is not clear partly from the language, so i have made some suggestions to insert the phrases in the comments or exchange some words that are crossed out with those in the comments (if u place your mouse cursor over the speech icon) in the attached pdf file . Hope the authors can check if the suggestions are what they meant

3. The formatting of tables in the original submission are better, except that the early and late infancy columns have been removed

4. Suggest that the 14% of the national population used in the calculation of the rates be displayed in the table instead of the whole national population birth data so that the denominator for the calculation of the SUDI rates are clear

5. More discussion on the limitations of the study due to the methodology can be added as given in the comments in the pdf

Overall, if these adjustments are made, the study would be useful to show some of the epidemiological findings of SUDI in japan

7. PLOS authors have the option to publish the peer review history of their article (what does this mean?). If published, this will include your full peer review and any attached files.

Reviewer #1: No

Reviewer #2: No

---

## [Author Response · Author response to Decision Letter 1]

3 Aug 2020

Reviewer #2: 

Thank you very much for the detailed comments to our submission. The raised comments were helpful to revise the manuscript. The reply is described below. The answer to miscellaneous comments in the PDF file is indicated in the file as well. Alterations are indicated in the revised text in red.

1. I would suggest to authors if removing the words "during sleep" and from the objective would make the study different? SUDI by definition happens during sleep. It would make the methodology wording less confusing

Reply

The authors agree with the point that the reviewer suggested. According to the comment, the words "during sleep" have been replaced by "sleep-related" as the first submission was.

2. The methodology is not clear partly from the language, so i have made some suggestions to insert the phrases in the comments or exchange some words that are crossed out with those in the comments (if u place your mouse cursor over the speech icon) in the attached pdf file . Hope the authors can check if the suggestions are what they meant

Reply

Thank you for the detailed check to the text. We followed all the suggestions. Please see the revised version, and the PDF file.

3. The formatting of tables in the original submission are better, except that the early and late infancy columns have been removed

Reply

According to the suggestion, all Tables 1 to 3 are formatted like the original version, except for the removal of early and late columns. 

4. Suggest that the 14% of the national population used in the calculation of the rates be displayed in the table instead of the whole national population birth data so that the denominator for the calculation of the SUDI rates are clear

Reply

The authors agree with what the reviewer pointed out. But the replacement in the column to the local population by multiplying 0.14 is unnatural for us. It is because only the total population is real. The area covered by the participants often overlaps to that of another institution. Each institution has a limited capacity for autopsy, so that police usually can choose the alternative one. The number of 0.14 is not strict at all, but actually tentative. We would like to show the total number as the last version was. Instead, a proviso is added into the bottom of the revised table with asterisk. Please see line 177.

5. More discussion on the limitations of the study due to the methodology can be added as given in the comments in the pdf

Reply

According to the advice, the statement concerning the limitations is transferred into the section of Discussion. Please see lines 294-295 and 310-312.

---

## [Editor Report · Decision Letter 2]

7 Aug 2020

Circumstances and factors of sleep-related sudden infant deaths in Japan

PONE-D-20-12699R2

Dear Dr. Osawa,

We’re pleased to inform you that your manuscript has been judged scientifically suitable for publication and will be formally accepted for publication once it meets all outstanding technical requirements.

Kind regards,

Ju Lee Oei

Academic Editor

PLOS ONE

---

## [Editor Report · Acceptance letter]

12 Aug 2020

PONE-D-20-12699R2 

Circumstances and factors of sleep-related sudden infancy deaths in Japan 

Dear Dr. Osawa:

I'm pleased to inform you that your manuscript has been deemed suitable for publication in PLOS ONE. Congratulations! Your manuscript is now with our production department. 

Kind regards, 

on behalf of

Dr. Ju Lee Oei 

Academic Editor

PLOS ONE